# Optimal intrusion detection for imbalanced data using Bagging method with deep neural network optimized by flower pollination algorithm

Hussein Ridha Sayegh[1,*], Wang Dong[1], Bahaa Hussein Taher[1], Muhanad Mohammed Kadum[2] and Ali Mansour Al-madani[1,*]

[1] College of Computer Science and Electronic Engineering, Hunan University, Changsha, Hunan, China
[2] School of Computer Science and Engineering, Central South University, Changsha, Hunan, China
* These authors contributed equally to this work.

## ABSTRACT

As the number of connected devices and Internet of Things (IoT) devices grows, it is becoming more and more important to develop efficient security mechanisms to manage risks and vulnerabilities in IoT networks. Intrusion detection systems (IDSs) have been developed and implemented in IoT networks to discern between regular network traffic and potential malicious attacks. This article proposes a new IDS based on a hybrid method of metaheuristic and deep learning techniques, namely, the flower pollination algorithm (FPA) and deep neural network (DNN), with an ensemble learning paradigm. To handle the problem of imbalance class distribution in intrusion datasets, a roughly-balanced (RB) Bagging strategy is utilized, where DNN models trained by FPA on a cost-sensitive fitness function are used as base learners. The RB Bagging strategy derives multiple RB training subsets from the original dataset and proper class weights are incorporated into the fitness function to attain unbiased DNN models. The performance of our IDS is evaluated using four commonly utilized public datasets, NSL-KDD, UNSW NB-15, CIC-IDS-2017, and BoT-IoT, in terms of different metrics, *i.e.*, accuracy, precision, recall, and F1-score. The results demonstrate that our IDS outperforms existing ones in accurately detecting network intrusions with effective handling of class imbalance problem.

## INTRODUCTION

The current evolution of Internet of Things (IoT) networks has led to unrestricted access to information and unpredictable behavior on the network (*Thakkar & Lohiya, 2021b*). Numerous network and Web applications are being used more often, which has led to the development of large amounts of data and increased dangers to the network environment. Moreover, a number of security risks and difficulties have been brought about by the growing use of IoT devices and the inherent architecture of IoT networks.



Corresponding authors
Hussein Ridha Sayegh,
husseinsayegh81@hnu.edu.cn
Ali Mansour Al-madani,
ali.m.almadani1992@gmail.com

To classify network data samples into attack and regular traffic in IoT networks, intrusion detection systems (IDSs) have been designed with a variety of approaches. Various machine learning (ML) techniques were employed to design new IDSs. For example, the implementation of the decision tree algorithm (*Modi et al., 2012*; *Peng et al., 2018*), support vector machine (SVM) models (*Wei et al., 2020*; *Schueller et al., 2018*), k-means (*Zhao & Zhang, 2016*; *Kumar, Mangathayaru & Narasimha, 2015*), k-nearest neighbor (k-NN) (*Ghosh, Mandal & Kumar, 2015*; *Deshpande et al., 2018*), and many other ML techniques (*Modi et al., 2013*; *Liu & Lang, 2019*; *da Costa et al., 2019*), were used for IDS design.

Motivated by their advantages over shallow learning methods, deep learning (DL) techniques with layered design have been recently employed in the development of new effective IDSs. The design of IDSs have been proposed using convolutional neural network (CNN) (*Wu, Chen & Li, 2018*), deep neural network (DNN) (*Thakkar & Lohiya, 2023*), deep recurrent neural network (deep RNN) (*Almiani et al., 2020*), restricted Boltzmann machines (RBMs) (*Dawoud, Shahristani & Raun, 2018*), multilayered perceptron neural network (MPNN) (*Hodo et al., 2016*), long short term memory (LSTM) (*Sayegh, Dong & Al-madani, 2024*), and many others (*Alkadi et al., 2021*). DL techniques are promise when trained with large amounts of evenly distributed data across class output labels (*Dong & Wang, 2016*). However, if there is a large disparity in the sample distribution among class output labels or if the training data samples are insufficient for learning, the performance of DL approaches is compromised (*Aminanto & Kim, 2016*). In terms of intrusion detection and classification, class imbalance may have an impact on performance, leading to a high rate of false positives and a low detection capabilities (*Liu et al., 2021*). This is because a system designed for intrusion detection and classification becomes biased when there is an unequal distribution of classes (*Thakkar & Lohiya, 2021c*), where such a system tends to learn more from the majority class and not enough from the minority class.

Various attempts have been proposed to address the class imbalance problem using data-level techniques that tend to balance the training data set either by creating new data samples from the minority class (over-sampling) or by eliminating some data samples from the majority class (under-sampling). The most common datal-level techniques are synthetic minority over-sampling (SMOTE), random over-sampling (ROS), and random under-sampling (RUS) (*Leevy et al., 2018*; *Sayegh, Dong & Al-madani, 2024*). Recently, a hybrid data resampling algorithm has been proposed in *Abdelkhalek & Mashaly (2023)*, which consists of over-sampling using adaptive synthetic sampling (ADASYN) and under-sampling using Tome-kLink. Further, *Sun, Du & Xiong (2024)* has proposed density-based under-sampling by applying different degrees of under-sampling to majority class samples across segmented regions. Data-level techniques, however, suffer from inherent drawbacks that could affect the process of learning and classification. On the one hand, over-sampling techniques can lead to overfitting, which tends to occur when duplicating existing minority class samples, or to class overlap, which tends to occur when the synthetic samples misrepresent the minority class. On the other hand, significant data samples from the majority class can be lost with under-sampling techniques.

Alternatively, some researchers have attempted to address the class imbalance problem using algorithm-level techniques that aim to increase the learning capacity of classification model with respect to the minority class. In general, algorithm-level techniques can be categorized into cost-sensitive and ensemble learning methods. Cost-sensitive methods assign a higher cost (weight) to the class of more interest (usually, the minority class) to enhance its detection accuracy (*Wu, Chen & Li, 2018*; *Mulyanto et al., 2021*). In ensemble learning, multiple learners (referred to as base learners) are trained and combined *via* some strategies, *e.g.*, Bagging and Boosting, to create a strong learner with generalization ability much better than that of an individual learner (*Thockchom, Singh & Nandi, 2023*). However, to have an effective ensemble, the base learners must be as good and independent as possible (*Zhou & Zhou, 2021*). Therefore, generating accurate and diverse base learners is the main challenge in ensemble learning with class imbalance datasets. To mitigate this challenge with intrusion detection datasets, researchers have recently resorted to hybridize ensemble learning with resampling techniques, *e.g.*, ROS (*Gupta, Jindal & Bedi, 2022*), SVM-SMOTE (*Gupta, Jindal & Bedi, 2022*), and dynamic distribution-based under-sampling (*Ren et al., 2023*), cost-sensitive DL (*Gupta, Jindal & Bedi, 2022*; *Thakkar & Lohiya, 2023*), and a combination of metaheuristic-based space partitioning and SMOTE (*Kamro, Rafiee & Mirjalili, 2024*).

The main purpose of this article is to evaluate the use of a metaheuristic optimization-based deep neural network (MHO-DNN) as a base learner in an ensemble learning paradigm to address the class imbalance in intrusion detection datasets. The term "MHO-DNN" is used to refer to hybridizing a metaheuristic optimization algorithm into a DNN in order to attain optimal architecture, optimal hyper-parameters, optimal weights and biases (*i.e.*, training DNN), and optimal feature representation level. In intrusion detection problem, it has been observed that such hybridization results in an improved performance of the IDS model, and can address problems such as diverse data distribution, and continuously changing environmental conditions, and huge network traffic (*Thakkar & Lohiya, 2020*). In this work, multiple DNN models are independently trained on various subsets of a given dataset using flower pollination algorithm (FPA), which is a metaheuristic inspired by the natural phenomenon of flower pollination, with a cost-sensitive objective function formulated by assigning proper class weights. Unlike traditional training algorithms, *i.e.*, gradient-based, that are easily trapped in local optima, FPA has the potential to provide global optimal solution as equipped with an escape mechanism from local optima entrapment. The reasons behind choosing FPA as an optimizer for DNN are the advantages of a fast convergence that is an exponentially better than that of other bio-inspired techniques as well as an excellent exploration and exploitation capabilities (*Thakkar & Lohiya, 2020*). The optimal DNN models are then used as base learners and combined to constitute an effective ensemble classifier *via* a roughly-balanced (RB) Bagging strategy (*Hido, Kashima & Takahashi, 2009*). To the best of our knowledge, our work is the first to combine a cost-sensitive MHO-DNN with ensemble learning for intrusion detection and classification.

The main contributions of this article are manifolded.

1) We propose a novel hybridization of a cost-sensitive MHO-DNN, namely, a DNN model trained based on FPA and a cost-sensitive objective function, with ensemble learning to handle the class imbalance problem in intrusion detection datasets. The use of a cost-sensitive objective function enhances the learning capacity of the individual DNN models with regard to the minority class. The global search capability of FPA enables accurate and diverse base learners required for an effective ensemble learning. The ensemble learning alleviates the over-predicting toward the minority class and rectifies for the individual learners' misleading prediction, and hence improves the generalization performance of the IDS for unseen data.

2) We evaluate the performance of the proposed method based on four intrusion detection datasets that are commonly used in the literature, NSL-KDD, UNSW NB-15, CIC-IDS-2017, and BoT-IoT data sets, in terms of accuracy, precision, recall, and F1-score.

3) We compare the performance of the proposed method with that of several state-of-the-art class imbalance methods. We also compare our model's performance with that of similar models trained based on popular metaheuristic algorithms, namely, genetic algorithm (GA), particle swarm optimization (PSO), and artificial bee colony (ABC).

The remainder of this article is structured as follows. The "Literature Review" section explores the related works for intrusion detection and classification. The proposed method is described in detail in "Methods". "Results and Discussion" presents the results and discussion regarding the performance evaluation and comparisons of the proposed method with other methods. Finally, the "Conclusions" section concludes the article.

## LITERATURE REVIEW

In the literature, an extensive research has been devoted to the design of coherent and potent IDS using ML and DL techniques. The main challenge lies in the imbalanced class distribution in intrusion detection datasets that hinders the performance of the classification model. Different techniques have been proposed to handle the class imbalance problem, such as LSTM-SMOTE (*Sayegh, Dong & Al-madani, 2024*), ROS (*Leevy et al., 2018*), RUS (*Leevy et al., 2018*), cost-sensitive learning (*Wu, Chen & Li, 2018*; *Mulyanto et al., 2021*), and ensemble learning (*Leevy et al., 2018*). In particular, focal loss function that involves a hyper-parameter balancing the importance of positive and negative samples was used in *Mulyanto et al. (2021)*.

Furthermore, various combinations of ensemble learning with other techniques have been proposed to address the problem of class imbalance in intrusion detection datasets. For example, ensemble learning has been combined with ROS, SVM-SMOTE, and cost-sensitive DL to design a three-layer IDS (*Gupta, Jindal & Bedi, 2022*). The developed IDS has been evaluated on NSL-KDD, CIDDS-001, and CICIDS2017 datasets, where it has shown a high attack detection rate and a small number of false alarms. *Thakkar & Lohiya (2023)* has also combined DNN with exactly-balanced (EB) Bagging ensemble learning (DNN-EB Bagging) in order to enhance the performance of IDS for IoT networks, which

has been evaluated on NSL-KDD, UNSW-NB 15, CIC-IDS-2017, and BoT-IoT. In *Ren et al. (2023)*, an IDS based on hybrid ensemble learning has been developed by incorporating dynamic distribution-based under-sampling into Boosting strategy. The evaluation of this IDS has been carried out using NSL-KDD and UNSW-NB15 datasets, demonstrating its high detection accuracy and low computational cost. *Kamro, Rafiee & Mirjalili (2024)* has attempted to eliminate class imbalance problem while minimizing alterations to the original data by combining SMOTE, metaheuristic-driven space partitioning, and ensemble learning.

On the other hand, the use of metaheuristic techniques to solve different optimization problems in the development of ML and DL models, *e.g.*, architecture optimization, hyperparameter optimization, training and feature representation level optimization, has been considerably increased in recent years (*Akay, Karaboga & Akay, 2022*). In intrusion detection problem, feature selection has been performed using GA (*Nguyen & Kim, 2020*), modified grey wolf optimizer (GWO) with principal component analysis (PCA) (*Swarna Priya et al., 2020*), and crow search algorithm (CSA) with opposition based learning (*SaiSindhuTheja & Shyam, 2021*). The authors in *Jayasankar, Kiruba Buri & Maheswaravenkatesh (2024)* have proposed an IoT environment-based IDS using dynamic search fireworks optimization–based feature selection with deep recurrent neural network. The proposed model efficiently detects the intrusions with an accuracy of 96.11%. In *Sanju (2023)*, an IDS model has been developed based on a modified metaheuristics algorithm with an ensemble of recurrent neural networks (RNNs), where feature selections were performed using harris hawk optimization (HHO) and fractional derivative mutation. This IDS has been evaluated on IoT-23, UNSW-NB15, and CICIDS2017. An optimized isolation forest (OIF)-based IDS has been proposed in *Elsaid & Binbusayyis (2024)*, where a modified version of HHO was utilized to reduce the dataset dimensionality by deleting irrelevant features. The proposed HHO-OIF-based IDS was evaluated on CICIDS-2018, NSL-KDD, and UNSW-NB15. Ant lion optimizer based feature subset selection with hybrid deep learning (CNN+LSTM) whose hyperparameters were optimized by FPA have been used in designing a MHO-DL-based IDS with an accuracy of 99.31% and 99.55% on ToN-IoT and CICIDS-2017 datasets respectively (*Alamro et al., 2023*). In *Alsudani & Ghazikhani (2023)*, the penguin optimization algorithm has been used to optimize the size of the hidden unit of a LSTM network, and the MHO-LSTM model was shown to 98.8% on NSL-KDD dataset. In *Malibari et al. (2022)*, quantum behaved particle swarm optimization (QPSO) is employed to optimize the hyperparameters of deep wavelet neural network model for the detection and classification of intrusions in the secured smart environment.

As for training DL models, gradient-based training algorithms have several drawbacks such as premature convergence and vanishing (or exploding) gradients leading to stagnation (or overshoot) during weight updates. With gradient-based search, training may converge to saddle points or overshoot. Metaheuristic techniques have been shown to overcome these drawbacks of gradient-based algorithms. GA, PSO, and ABC are the most common metaheuristics used for training of DL models in designing IDS (*Thakkar & Lohiya, 2020*). Recently, chronological salp swarm algorithm was employed to obtain the

optimal weights for deep belief network (DBN) (*Karuppusamy et al., 2022*), and the accuracy of the proposed MHO-DBN-based IDS is 96.18% and 97.64% on KDD cup and BoT-IoT datasets respectively. *Kayyidavazhiyil (2023)* combined classification using deep metaheuristics artificial neural network and feature selection based on enhanced genetic sine swarm intelligence to develop an IDS that has been shown to perform well on UNSW-NB15 and NSL-KDD datasets. In *Hacılar et al. (2024)*, an IDS has been proposed using a deep autoencoder-based, vectorized, and parallelized ABC algorithm for training feed-forward artificial neural networks, and was evaluated on UNSW-NB15 and NF-UNSW-NB15-v2 datasets. In UNSW-NB15 dataset, the detection rate was increased by 0.76 to 0.81 and the false alarm rate was reduced by 0.016 to 0.005 compared to that based on the backpropagation algorithm. An MHO-ANN based IDS was developed in *Kumari et al. (2024)*, where spider monkey optimization was utilized to train ANN using the datasets Luflow, CIC-IDS 2017, UNR-IDD and NSL-KDD. The proposed model has accuracy of 100% and 99% on the binary Luflow dataset and the multiclass NSL-KDD dataset respectively.

Despite the growing attention towards MHO-DL in recent research, there is still a much room for further investigation with respect to different combinations of various metaheuristics and ANN types aiming at developing IDS of high detection accuracy and low computational burden with the capability to adapt to the continuously changing network scenarios, *e.g.*, intrusion activities and user actions. This research gap makes sense, given the wide variety of metaheuristics and the fact stated in the "No-Free Lunch" theorem, that is, no single metaheuristic consistently outperforms all others on all optimization problems.

In summary, MHO-ML/DL, cost-sensitive learning, and ensemble learning are promising techniques for classifying imbalanced datasets. Nevertheless, little attention has been paid to investigate the performance of the integration of cost-sensitive learning, MHO-DL, and ensemble learning in handling the class imbalance problems in intrusion detection and classification.

## METHODS

In this section, we describe the methods carried out to fulfill the purpose of this study. We start with a description of the training datasets, and then move to data preprocessing. Next, we present architecture of the utilized DNN. Then, we illustrate the proposed FPA-based training algorithm for the DNN model. Finally, the proposed Bagging-based ensemble learning is described.

### Datasets

Four intrusion detection datasets, namely NSL-KDD, UNSW NB-15, CIC-IDS-2017, and BoT-IoT, were used to evaluate the performance of the proposed IDS. Notice that the proposed IDS is evaluated on multiple datasets to manifest the wide intrusion detection capability of the proposed approach, where such datasets are created under various network circumstances and are made up of a variety of network characteristics.

The NSL-KDD dataset is a curated version of the KDD 99 dataset, addressing the limitations and challenges of the original version. There are 41 attributes in NSL-KDD representing the features of the data, three of which are categorical and the remaining features are numerical. There is also one attribute indicating the class of the record among the five classes, normal, denial-of-service (DoS) attack, probe attack, remote to local (R2L) attack, and user to root (U2R) attack. Although it fails to resemble true real-world networks, NSL-KDD dataset is a valuable benchmark for evaluating IDS due to its availability and refinements.

The UNSW-NB-15 dataset is an important and comprehensive repository tailored for the evaluation and advancement of IDS. It was developed at the UNSW cyber security lab using the IXIA PerfectStorm tool to address the limitations of existing datasets. A diverse and realistic collection of network traffic data is provided in UNSW-NB-15 by hybridizing the real modern normal behaviors and the synthetical attack activities. This dataset involves nine different attacks, backdoor, shellcode, reconnaissance, worms, fuzzers, DoS, generic, analysis, and exploits. It comprises 42 features, three of which are categorical and the remaining features are numeric.

The CIC-IDS-2017 dataset plays a crucial role in advancing IDS since it addresses key limitations previously established datasets by providing up-to-date and diverse network traffic data, including both benign and known attacks scenarios. It was developed by the Canadian Institute of Cyber Security by attack structures and normal traffic flows during a test period of 5 days. It consists of seven attack categories that represent the most up-to-date attack scenarios, including brute force attack, heart bleed attack, botnet, DoS attack, DDoS attack, web attack, and infiltration attack. In CIC-IDS-2017 dataset, there are 79 features that represent various metrics of network traffic.

The BoT-IoT dataset is a modernized and realistic dataset that created with a focus on IoT network. It was developed in the Cyber Range Lab of UNSW Canberra using a realistic testbed. It comprises both normal IoT-related and other network traffic, as well as traffic of a variety of attack categories that botnets often use. It contains four attack categories, DDoS, DoS, reconnaissance, and information theft attacks.

The characteristics of the four datasets are given in Table 1.

## Data preprocessing

Data preprocessing is a critical component for building a robust and accurate intrusion detection system. Encoding and normalization techniques are applied to transform the input data into a uniform format in order to feed it to DNN for further processing. That is, one-hot encoding method is employed to convert categorial features into numerical features (*Thakkar & Lohiya, 2021a*). Afterwards, standard scalar method is employed to normalize these features (*Thakkar & Lohiya, 2021a*).

Each dataset was split into a training dataset including 80% of the data and a test dataset comprising of the remaining 20%. Moreover, a 10-fold cross-validation approach was utilized during the training process in order to avoid the over-fitting problem.

**Table 1 Characteristics of the datasets.**

| Element | NSL-KDD | UNSW-NB 15 | CIC-IDS 2017 | BoT-IoT |
|---|---|---|---|---|
| Type of network traffic | Real and synthetic | Synthetic | Real | Real |
| Number of features | 41 | 42 | 79 | 15 |
| Number of attack categories | 4 | 9 | 7 | 4 |
| Number of classes | 5 | 10 | 15 | 5 |
| Number of data samples | 148,517 | 257,673 | 225,745 | 3,668,522 |
| Number of benign samples | 77,054 | 93,000 | 128,027 | 477 |
| Number of attack samples | 71,463 | 164,673 | 97,718 | 3,668,045 |
| Number of training samples | 118,814 | 206,138 | 180,596 | 2,934,817 |
| Number of test samples | 29,703 | 51,535 | 45,149 | 733,705 |

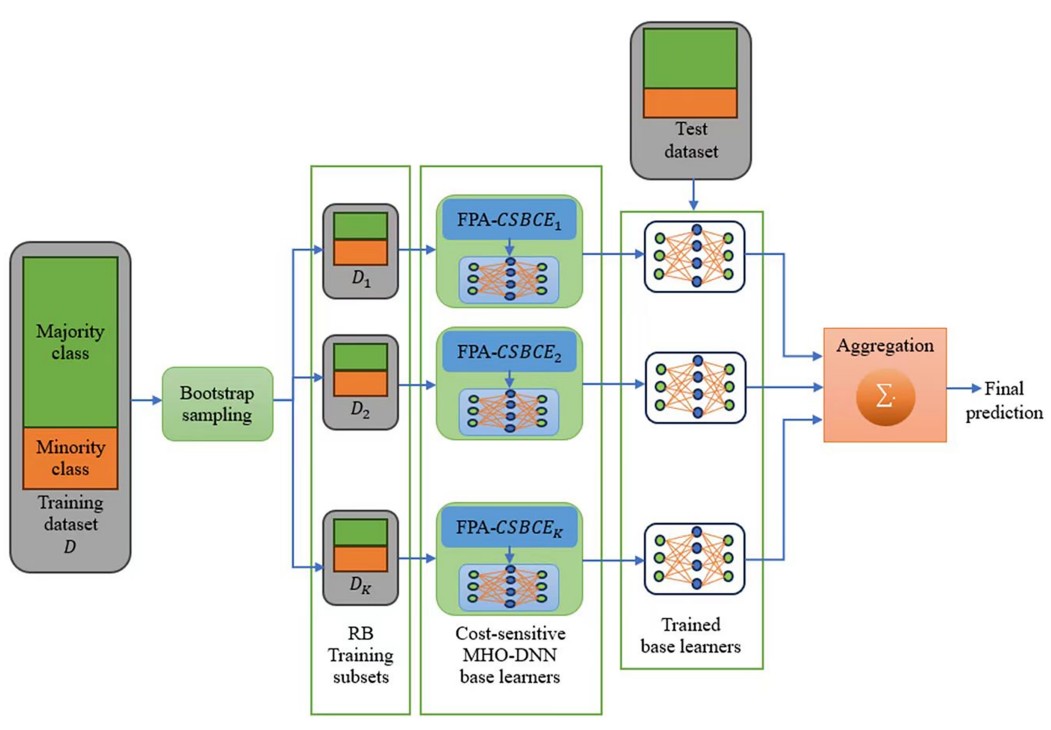

**Figure 1 The proposed DNN architecture.**

## DNN architecture

The DNN architecture used in this work is given in Fig. 1. As can be seen, the utilized DNN consists of an input layer, three hidden layers, and an output layer. Notice that we chose this number of hidden layers since it provides us a better tradeoff between the complexity of the model required to capture intricate details of data and maintaining the model as simple as possible for efficient performance. The size of the input layer is determined by the number of features in the dataset being used. Based on trial-and-error method, we chose the number of nodes in the hidden layers as 128, 64, and 32 for CIC-IDS-2017,

respectively, and 64, 32, and 16 for the other datasets, respectively. The activation function of the hidden layers is ReLU, which expedites learning and promotes faster convergence. We chose the Sigmoid activation function for the node of the output layer, since it is appropriate for binary classification and its non-linearity give the model ability to capture complex features in the data.

## The proposed metaheuristic-based training algorithm

The proposed training algorithm is based on a recently-proposed metaheuristic, the flower pollination algorithm (FPA). Despite the high availability of optimization algorithms in the relevant literature, we choose FPA as an optimizer for DNN due to the advantages of a fast convergence that is an exponentially better than that of other bio-inspired techniques as well as an excellent exploration and exploitation capabilities (*Thakkar & Lohiya, 2020*). In this section, we first review the theoretical framework of FPA. Next, we discuss the encoding of the search agents and the representation of the population in FPA. Then, we describe the fitness function used in the proposed training algorithm.

### *Flower pollination algorithm*

FPA is inspired by the pollination process in plans (*Yang, 2012*). The pollination process can be biotic and/or abiotic. It follows that, the mathematical modelling of FPA is formulated as follows.

1) The biotic process (which corresponds to global pollination) is modeled as:

$$x_i^{t+1} = x_i^t + \gamma L(\gamma)(g_* - x_i^t), \tag{1}$$

where $g_*$ denotes the global optimal solution obtained so far, $\gamma$ refers to a scaling factor that used to control the step size, and $x_i^t$ represents the $i$-th pollen, *i.e.*, the solution vector $x_i$ at the $t$-th iteration. $L(\gamma)$ denotes the Lévy factor that used to transfer the pollens between various flower species, which is computed by *Yang (2012)*:

$$L(\gamma) \sim \frac{\lambda \Gamma(\lambda) \sin\left(\frac{\pi \lambda}{2}\right)}{\pi} \frac{1}{h^{1+\lambda}}, \quad h \gg h_0 > 0, \tag{2}$$

where $h$ denotes the step size. The assigned step-size value $h_0$ is often chosen as small as 0.1 (notice that large values of $h_0$ are also acceptable).

2) The abiotic (which corresponds to local pollination) process is modeled as:

$$x_i^{t+1} = x_i^t + \varepsilon(x_j^t - x_k^t), \tag{3}$$

where $x_j^t$ and $x_k^t$ are different pollens of the same plant, and $\varepsilon$ is chosen from a uniform distribution $\in [0, 1]$.

3) To switch between the global and local pollination, a switching probability factor $p$ is selected in the interval $[0, 1]$.

The pseudo-code of FPA is summarized in Algorithm 1.

---

**Algorithm 1** Flower pollination algorithm.

Define $p$ (switching probability), $n$ (number of flowers or pollen gametes),

and *IterMax* (maximum number of iterations);

Initialize a population of $n$ random solutions $\{x_i, i = 1, \cdots, n\}$ (each search agent corresponds to a flower);

Compute the fitness values for the initialized population;

Find the best solution $g_*$ among the population (*i.e.*, the solution with the smallest fitness value);

**while** $t < IterMax$ **do**

    **for** $i = 1 : n$ **do**

        **if** *rand* $< p$ **then**

            Compute $L(\gamma)$ using Eq. (2);

            Perform global pollination using Eq. (1);

        **else**

            Draw $\varepsilon$ from $[0, 1]$;

            Perform local pollination using Eq. (3);

        **end if**

        Compute the fitness value for the new solution;

        **if** the new solution is better than the old solution **then**

            Update the solution in the population;

        **end if**

        Find the best solution $g_*$ in the new population;

    **end for**

    $t = t + 1$;

**end while**

return $g_*$;

---

### Encoding of flowers

Now, we present the encoding of the search agents (flowers) to represent the learning parameters of the DNN architecture shown in Fig. 1. The learning parameters are the connection weights between successive layers and the biases. Consider $I$ is the number of input features, and $m_1$, $m_2$, and $m_3$ are the number of neurons in the first, second, and third hidden layer, respectively. Then, the weights between the input layer and the first hidden layer are $w_{1,1}^{(1)}, \cdots, w_{I,m_1}^{(1)}$. The weights between the first hidden layer and the second hidden layer are $w_{1,1}^{(2)}, \cdots, w_{m_1,m_2}^{(2)}$. The weights between the second hidden layer and the third hidden layer are $w_{1,1}^{(3)}, \cdots, w_{m_2,m_3}^{(3)}$. The weights between the third hidden layer and the output layer are $w_1^{(o)}, \cdots, w_{m_3}^{(3)}$. The biases of the three hidden layers are

$$\beta_1^{(1)}, \cdots, \beta_{m_1}^{(1)}, \beta_1^{(2)} \cdots \beta_{m_2}^{(2)}, \text{ and } \beta_1^{(3)}, \cdots, \beta_{m_3}^{(3)}.$$

Therefore, the vector of the search agent can be represented by

$$
\begin{aligned}
x = \Big[ & w_{1,1}^{(1)} \cdots w_{I,m_1}^{(1)} \ w_{1,1}^{(2)} \cdots w_{m_1,m_2}^{(2)} \ w_{1,1}^{(3)} \cdots w_{m_2,m_3}^{(3)} \\
& w_1^{(o)} \cdots w_{m_3}^{(o)} \ \beta_1^{(1)} \cdots \beta_{m_1}^{(1)} \ \beta_1^{(2)} \cdots \beta_{m_2}^{(2)} \\
& \beta_1^{(3)} \cdots \beta_{m_3}^{(3)} \ \beta_1^{(o)} \Big].
\end{aligned}
\tag{4}
$$

It can be inferred that the number of elements in this vector is given by

$$
M = I \times m_1 + m_1 \times m_2 + m_2 \times m_3 + m_3 + m_1 + m_2 + m_3 + 1.
\tag{5}
$$

Then, the population of FPA can be represented by an $n \times M$ matrix, where each row represents a flower.

### Fitness function

The fitness function used in the training process of the proposed DNN is the binary cross entropy (BCE) function. Given a dataset with $N_k$ data samples, where $k$ denotes the index of the dataset, the BCE fitness function is defined by

$$
BCE_k = -\frac{1}{N_k} \left[ \sum_{i=0}^{N_k} \left[ y_i \log(\hat{y}_i) + (1 - y_i) \log(1 - \hat{y}_i) \right] \right],
\tag{6}
$$

where $y_i$ is the truth value for the $i$th data sample, and $\hat{y}_i$ is the predicted probability for the $i$th data sample. We chose this maximum-likelihood based loss function since it results in well-calibrated probabilities and mitigates the exponential characteristics of the activation function. The cost-sensitive BCE fitness function is defined by

$$
CSBCE_k = -\frac{1}{N_k} \left[ \sum_{i=0}^{N_k} \left[ W_k y_i \log(\hat{y}_i) + (1 - y_i) \log(1 - \hat{y}_i) \right] \right],
\tag{7}
$$

where $W_k$ is a weighting factor used to balance the learning process for both classes on the given dataset. This article aims to employ FPA to find the optimal values of the learning parameters of the DNN architecture to keep $CSBCE_k$ as minimum as possible. The optimal solution is obtained by

$$
\begin{aligned}
g_k^* = &\ argmin \ CSBCE_k \\
&\text{s.t. } (x_1, x_2, \ldots x_n) \in [-1, 1]^n,
\end{aligned}
\tag{8}
$$

where $x_1, x_2, \ldots x_n$ are the search agents of FPA. Note that a penalty function method has been used for handling the boundary constraints, which converts the constrained optimization problem into unconstrained problem by introducing a penalty for violating the constraints.

### Bagging-based ensemble learning

The term "ensemble learning" refers to the integration of multiple base learners *via* some strategies, *e.g.*, Bagging and Boosting, to create a strong learner with generalization ability

much better than that of an individual learner (*Zhou & Zhou, 2021*). In this work, we utilize an improved Bagging strategy to construct our ensemble models. In the original Bagging strategy (*Breiman, 1996*), multiple base learners are trained using bootstrapped data subsets formed by randomly drawing data instances from the original dataset regardless of the class labels, and the outputs of the base learners are aggregated to make the final predictions. Each training subset is derived with unique statistical properties, which results in diversity in learning process required for effective ensemble learning. The variation around the data instances in these subsets limits the noise brought on the statistical characteristics of data instances. Thus, Bagging strategy removes the potential for over-fitting seen in individual learners constructed with the full training dataset.

Nevertheless, the original Bagging strategy does not deal with the class imbalance problem directly, so it has to be modified or hybridized with another technique to handle this problem. Common methods, *i.e.*, exactly-balanced Bagging strategies, focused on deriving training subsets with exactly the same number of majority and minority samples so as the base learners are trained equally on both classes. However, this way can result in base learners that are less diverse (*Hido, Kashima & Takahashi, 2009*). In contrast, roughly-balanced (RB) Bagging strategy was proposed in *Hido, Kashima & Takahashi (2009)*, in which the number of minority samples is fixed to $N^{pos}$ (where $N^{pos}$ is the number of minority samples in the original dataset), whereas the number of majority samples drawn for each base learner varies slightly following a negative binomial distribution (with probability of success $q = 0.5$ and number of successes $n = N^{pos}$). Given an imbalanced dataset $D = (X_i, y_i) : X_i \in R^I, y_i \in 0, 1$, where $I$ is the number of the features, the minority set $D^{pos}$ and majority set $D^{neg}$ are defined by

$$D^{pos} = (X_i, y_i) : X_i \in R^P, y_i = 1, \quad D^{neg} = (X_i, y_i) : X_i \in R^p, y_i = 0. \tag{9}$$

For $K$ base learners, $D$ is used to create $K$ RB training subsets $D_k = D^{pos} \cup D_k^{neg}$, $k = 1, \cdots, K$, where $|D_k^{neg}|$ is selected based on a negative binomial distribution. The pseudo-code of the RB Bagging strategy given in Algorithm 2. It should be noted that in this strategy the class distribution of the training subsets becomes slightly imbalanced (just like the original Bagging strategy for balanced datasets). In fact, the fluctuation within the class ratios of the training subsets in the RB Bagging strategy was shown to increase the diversity compared to exactly-balanced Bagging strategies.

To maintain effective class representation during the bagging process, our strategy carefully controls the attack proportion distribution across base learners. While all minority (attack) samples $N_{pos}$ are maintained in each subset to preserve attack patterns, the majority class samples $N_{neg}^k$ are sampled following a negative binomial distribution with probability $q = 0.5$ and number of successes $n = N_{pos}$. This approach ensures that:

- Each base learner maintains sufficient exposure to attack patterns
- The controlled variation in majority class sampling promotes diversity among base learners
- The class weights $W_k = N_{neg}^k / N_{pos}$ dynamically adjust for the specific imbalance in each subset

---

**Algorithm 2** The RB Bagging algorithm.

*Input:*

        $D$ is the training dataset

        $L$ is the algorithm for the base learners

        $K$ is the number of base learners

        $x_i$ is an example drawn from the training set;

*Build RB bagging model(D,L,K):*

        Divide $D$ into $D^{pos}$ and $D^{neg}$;

        Set $N^{pos}$ as the size of $D^{pos}$;

     **for** $k = 1$ to $K$ **do**

        Draw $N^{neg}$ from the negative binomial distribution with $n = N^{pos}$ and $q = 0.5$;

        Let $D_k^{neg}$ be $N_k^{neg}$ samples from $D^{neg}$ with or without replacement;

        Let $D_k^{pos}$ be $N^{pos}$ samples from $D^{pos}$ with or without replacement;

        Build a base learner model $f^k(x)$ by $L$ based on $D^{pos} \cup D_k^{neg}$;

   **end for**

        Combine all $f^k(x)$ into the aggregated model $f^A(x)$;

        Return $f^A(x)$;

*Predict(f(x), x_i, y):*

        Calculate $p^A(y|x_i) = \frac{1}{K}\sum_{k=1}^{K} p^k(y|x_i)$ for all $y$;

        Let $\hat{y} = \mathrm{argmax}\, p^A(y|x_i)$;

        Return $\hat{y}$;

---

This sampling strategy helps prevent over-dilution of attack patterns while maintaining the beneficial effects of ensemble diversity.

In this article, the RB Bagging strategy is utilized, where the base learners are cost-sensitive FPA-DNN models. In addition, to have base learners that perform equally for both classes, we assign appropriate class weights for the respective fitness functions. That is, for the $k$th base learner, we use the weight,

$$W_k = \frac{N_k^{neg}}{N^{pos}}, \tag{10}$$

where $N_k^{neg}$ is the number of majority samples in $D_k^{neg}$. The schematic of the proposed IDS is shown in Fig. 2. As shown in this figure, $K$ of RB training subsets are created from the original imbalanced training dataset, and each subset is fed into a cost-sensitive FPA-DNN base learner for training purposes. Note that in our experiments the value of $K$ is set to the default value in Bagging strategy, *i.e.*, $K = 10$ (*Kadiyala & Kumar, 2018*). Due to the creation of distinct training subsets for each base learner, the cost-sensitive FPA-DNN models perform better and can generalize more effectively for intrusion detection and classification.

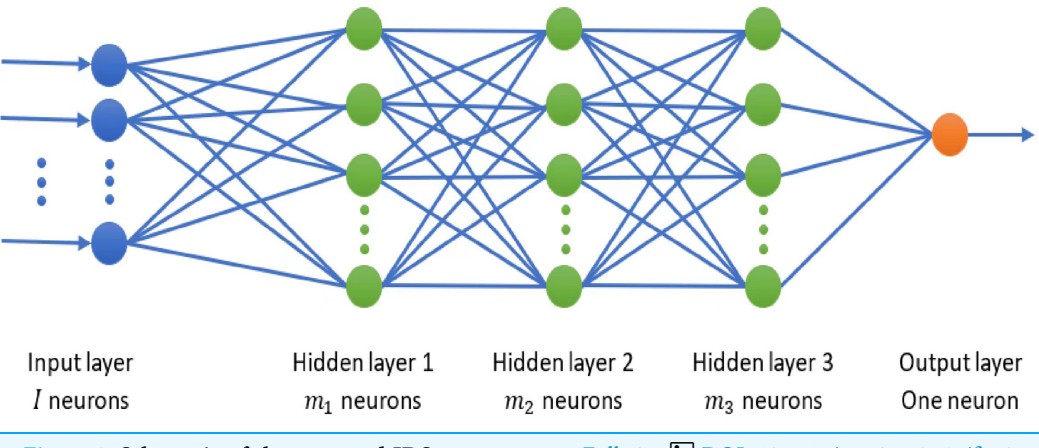

**Figure 2 Schematic of the proposed IDS.**

## Evaluation metrics

The performance of the proposed IDS is evaluated on the four widely-used datasets: NSL-KDD, UNSW NB-15, CIC-IDS-2017, and BoT-IoT. The following evaluation metrics are used:

$$Accuracy = \frac{TP + TN}{TP + TN + FP + FN}, \tag{11}$$

$$Precision = \frac{TP}{TP + FP}, \tag{12}$$

$$Recall = \frac{TP}{TP + FN}, \tag{13}$$

$$F1 - score = 2 \cdot \frac{Precision \cdot Recall}{Precision + Recall}, \tag{14}$$

where $TP$ (true positive), $FN$ (false negative), and $FP$ (false positive) represent the total number of truly detected, undetected, and wrongly detected attacks, respectively. Whereas, $TN$ (true negative) represents the total number of truly prediction of benign.

## RESULTS AND DISCUSSION

Extensive experiments were conducted to evaluate the performance of the proposed IDS developed based on a hybridization of cost-sensitive FPA-DNN with RB Bagging strategy on the datasets: NSL-KDD, UNSW NB-15, CIC-IDS-2017, and BoT-IoT. To have a precise evaluation, the outcomes for the proposed IDS were acquired for 30 independent runs on each dataset, and the averaged values for the accuracy, precision, recall, and F1-score evaluation metrics were computed. For comparison reasons, equivalent experiments were conducted for similar IDSs developed based on widely-known metaheuristics, *i.e.*, GA, PSO, and ABC. The control parameters of the employed metaheuristics are presented in Table 2. The performance of our IDS is also compared to that of recently-proposed IDSs developed based on data-level, cost-sensitive, ensemble learning, and MHO-ML methods.

**Table 2  Control parameters of the employed metaheuristics.**

| Parameter | GA | PSO | ABC | FPA |
|---|---|---|---|---|
| Population size ($n$) | 50 | 50 | 50 | 50 |
| Maximum iterations (*IterMax*) | 5,000 | 5,000 | 5,000 | 5,000 |
| Crossover function | Two-point | – | – | – |
| Mutation rate | 0.1 | – | – | |
| Selection method | Roulette wheel | – | – | – |
| Cognitive constant | – | 2.0 | – | – |
| Social constant | – | 2.0 | – | – |
| Initial and final velocity | – | 0.05 and 1 | – | – |
| Minimum and maximum inertia weights | – | 0.4 and 0.9 | – | – |
| Number of employed bees | – | – | 8 | – |
| Switching probability $p$ | – | – | – | 0.85 |

From existing data-level methods, we chose our proposed LSTM-SMOTE IDS (*Sayegh, Dong & Al-madani, 2024*). Focal loss-based IDS was chosen as a cost-sensitive method (*Mulyanto et al., 2021*). We chose the ensemble learning-based IDS proposed in *Thakkar & Lohiya (2023)*, which is developed based on DNN and EB Bagging strategy (DNN-EB Bagging). Recent studies have demonstrated the significance of handling extremely imbalanced datasets in intrusion detection. For instance, some advanced persistent threat (APT) detection scenarios deal with attack ratios as low as 0.04% of total traffic (*Benabderrahmane et al., 2024*), presenting even greater challenges than traditional IDS datasets. Such extreme imbalances require particularly robust detection approaches, making the evaluation of IDS methods against heavily imbalanced data increasingly important. The hybrid HHO-OIF based IDS was chosen from existing MHO-ML/DL methods (*Elsaid & Binbusayyis, 2024*), in which feature selection is carried out using HHO. Notice that all the experiments were carried out using Python on Intel Core i7 CPU with a 64-bit Windows 11 operating system and 8.00 GB RAM.

First, the performance of the proposed IDS for an arbitrary run from the thirty independent runs is investigated. The confusion matrices corresponding to this run are given in Fig. 3. A confusion matrix summarizes the count of *TP*, *FP*, *TN*, and *FN* predictions, which allows to derive various evaluation metrics, including accuracy, precision, recall, and F1-score. Moreover, such a matrix provides a comprehensive understanding of the model's capabilities, identifying its strengths, and highlighting potential areas for improvement. As can be seen from Fig. 3, only 24, 85, 38, and 11 attacks out of 16,604, 33,679, 25,049, and 733,623 were not detected in NLS-KDD, UNSW-NB15, CIC-IDS 2017, and BoT-IoT datasets respectively. Thus, our IDS achieves high detection rates in the four datasets. Further, the loss curves corresponding to the chosen run are illustrated in Fig. 4, indicating the convergence of our IDS during training process. At the initial training phase, the curves exhibit a significant downward trend as the model's performance is continuously improved. After 60–80% of epochs, the curves tend to be flat

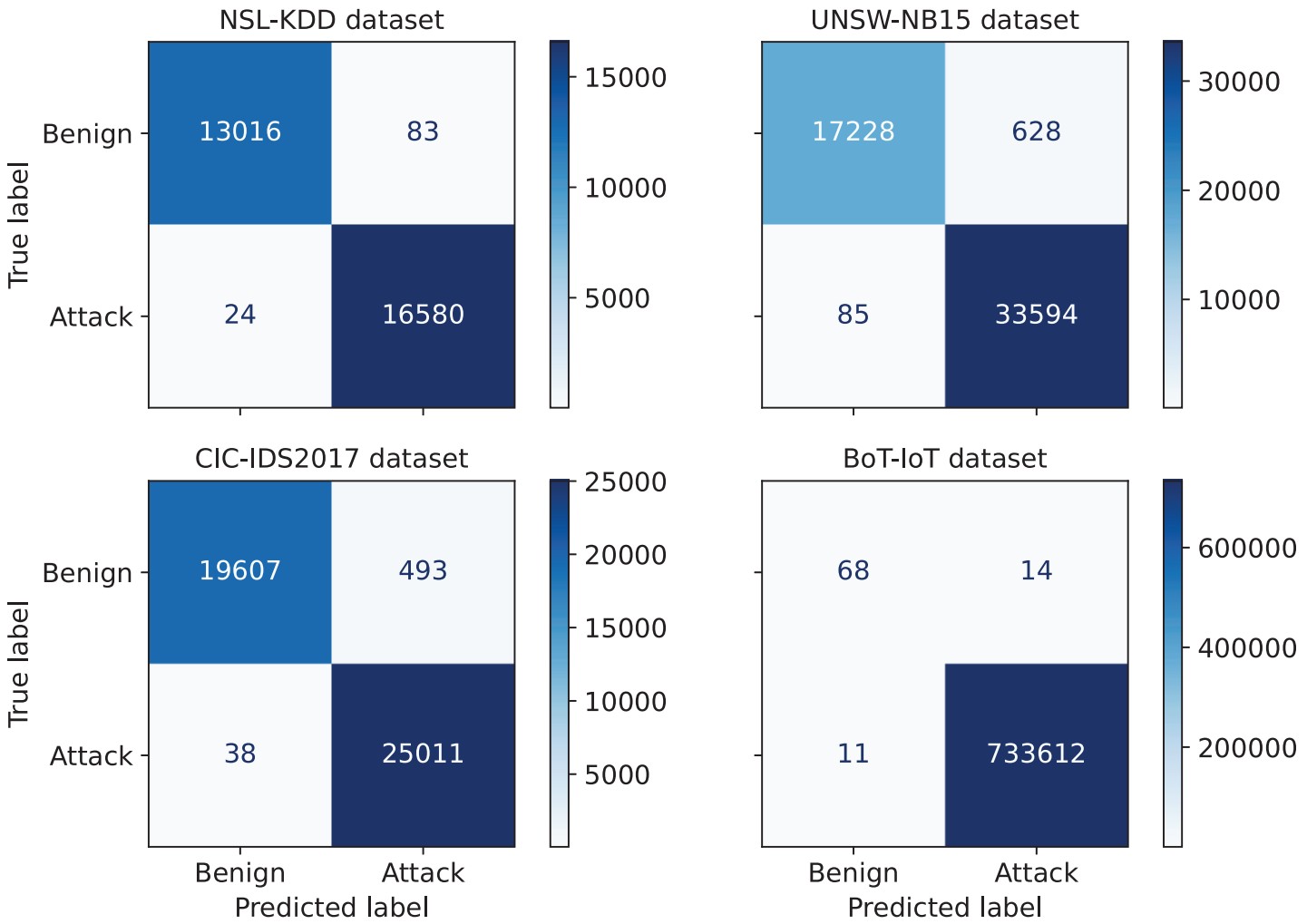

**Figure 3** Confusion matrices for the proposed IDS on the NSL-KDD, UNSW-NB15, CIC-IDS 2017, and BoT-IoT datasets.

as the performance becomes more stable. Notice that the early termination of training guarantees the prevention of over-fitting.

Second, the comparison results for the considered IDSs in terms of the accuracy, precision, recall, and F1-score metrics are presented in Tables 3–6, respectively (the percentage between parentheses corresponds to the respective improvement percentage over the benchmark model.). As can be seen from Table 3, our IDS achieves the highest accuracy for all the four datasets, with 99.59% for NSL-KDD, 98.33% for UNSW NB-15, 99.32% for CIC-IDS-2017, and 99.41% for BoT-IoT. When comparing our model with the recently-proposed ones, an improvement in the accuracy of about 1% over the best model is observed. Similarly, when comparing our FPA-based IDS with other metaheuristics-based IDSs, we notice an improvement in the accuracy of order of about 2% to 5% over the ABC-based IDS.

However, in imbalanced data distribution scenarios, the accuracy metric alone is insufficient to identify the performance with respect to the minority class. The precision

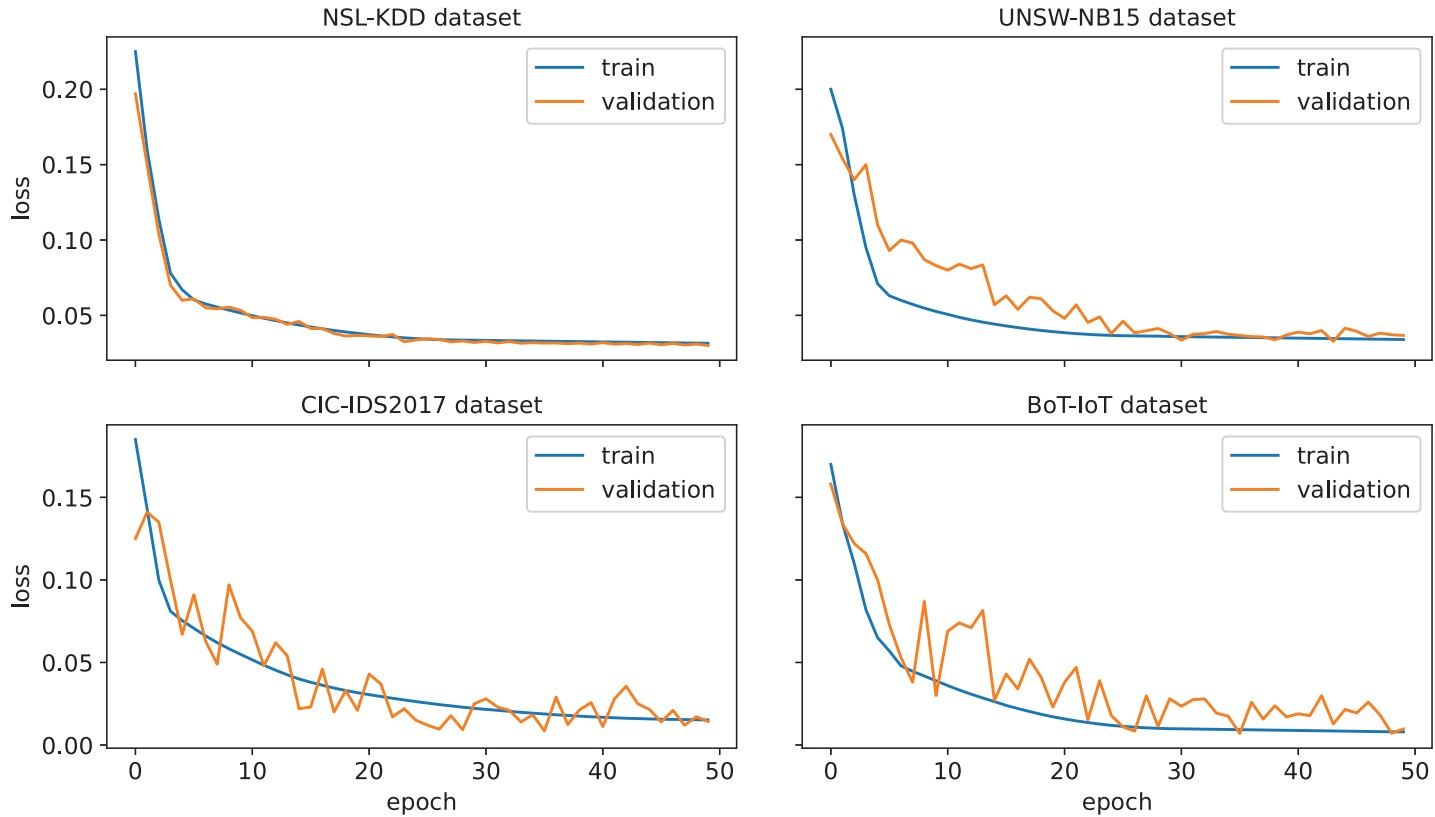

**Figure 4 Loss function for the proposed IDS on the NSL-KDD, UNSW-NB 15, CIC-IDS 2017, and BoT-IoT datasets.**

**Table 3 Comparison for the average accuracy of the considered IDSs.**

| Method | NSL-KDD | UNSW-NB 15 | CIC-IDS 2017 | BoT-IoT |
|---|---|---|---|---|
| LSTM-SMOTE | 98.22 (1.40%) | 96.93 (1.44%) | 98.87 (0.46%) | 98.00 (1.44%) |
| Focal loss | 83.95 (18.63%) | 86.73 (13.38%) | 94.28 (5.35%) | 75.29 (32.04%) |
| DNN-EB Bagging | 97.25 (2.41%) | 96.08 (2.34%) | 98.29 (1.05%) | 96.02 (3.53%) |
| HHO-OIF | 95.6 (4.17%) | 94.8 (3.72%) | 95.45 (4.05%) | 96.11 (3.43%) |
| cost-sensitive GA-DNN-RB Bagging | 96.68 (3.01%) | 94.57 (3.98%) | 92.56 (7.30%) | 92.06 (7.98%) |
| cost-sensitive PSO-DNN-RB Bagging | 97.39 (2.26%) | 95.72 (2.73%) | 94.78 (4.79%) | 94.04 (5.71%) |
| cost-sensitive ABC-DNN-RB Bagging | 97.77 (1.86%) | 96.56 (1.83%) | 95.57 (3.92%) | 94.55 (5.14%) |
| Proposed | 99.59 | 98.33 | 99.32 | 99.41 |

and recall evaluation metrics are also presented to demonstrate the efficacy of our IDS. It should be pointed out that, in our classification problem, low value of the precision metric indicates a high number of wrongly detected attacks, and low value of the recall metric indicates a high number of undetected attacks. From Tables 4 and 5, the precision and recall metrics for our IDS are the highest for all the four datasets, where they lie within 98.38–99.99% and 99.02–99.90%, respectively. When comparing our IDS with the recently-proposed and metaheuristics-based models, we observe that no other model

**Table 4 Comparison for the average precision of the considered IDSs.**

| Method | NSL-KDD | UNSW-NB 15 | CIC-IDS 2017 | BoT-IoT |
|---|---|---|---|---|
| LSTM-SMOTE | 97.10 (2.23%) | 96.24 (2.22%) | 95.23 (3.49%) | 97.91 (2.12%) |
| Focal loss | 84.47 (17.52%) | 90.66 (8.52%) | 97.48 (1.10%) | 83.99 (19.05%) |
| DNN-EB Bagging | 95.39 (4.07%) | 95.37 (3.16%) | 93.05 (5.91%) | 95.93 (4.23%) |
| HHO-OIF | 98.28 (1.01%) | 98.31 (0.07%) | 97.30 (1.28%) | 97.12 (2.96%) |
| cost-sensitive GA-DNN-RB Bagging | 95.00 (4.49%) | 93.17 (5.59%) | 73.38 (34.30%) | 91.98 (8.71%) |
| cost-sensitive PSO-DNN-RB Bagging | 95.72 (3.71%) | 94.29 (4.34%) | 79.94 (23.28%) | 93.96 (6.42%) |
| cost-sensitive ABC-DNN-RB Bagging | 96.14 (3.26%) | 95.44 (3.08%) | 82.41 (19.59%) | 94.45 (5.87%) |
| Proposed | 99.27 | 98.38 | 98.55 | 99.99 |

**Table 5 Comparison for the average recall of the considered IDSs.**

| Method | NSL-KDD | UNSW-NB 15 | CIC-IDS 2017 | BoT-IoT |
|---|---|---|---|---|
| LSTM-SMOTE | 99.41 (0.49%) | 97.68 (1.37%) | 99.20 (0.71%) | 90.39 (9.98%) |
| Focal loss | 85.02 (17.50%) | 90.82 (9.03%) | 94.70 (5.49%) | 80.90 (22.88%) |
| DNN-EB Bagging | 99.30 (0.60%) | 96.87 (2.22%) | 98.66 (1.26%) | 88.56 (12.25%) |
| HHO-OIF | 95.53 (4.57%) | 94.39 (4.91%) | 96.47 (3.56%) | 95.34 (4.27%) |
| cost-sensitive GA-DNN-RB Bagging | 98.55 (1.37%) | 96.20 (2.93%) | 97.60 (2.36%) | 84.91 (17.08%) |
| cost-sensitive PSO-DNN-RB Bagging | 99.21 (0.70%) | 97.34 (1.73%) | 98.11 (1.82%) | 86.74 (14.61%) |
| cost-sensitive ABC-DNN-RB Bagging | 99.54 (0.36%) | 97.78 (1.27%) | 98.54 (1.38%) | 87.19 (14.02%) |
| Proposed | 99.90 | 99.02 | 99.90 | 99.41 |

**Table 6 Comparison for the average F1-score of the considered IDSs.**

| Method | NSL-KDD | UNSW-NB 15 | CIC-IDS 2017 | BoT-IoT |
|---|---|---|---|---|
| LSTM-SMOTE | 98.24 (1.36%) | 96.95 (1.80%) | 97.19 (2.09%) | 94.00 (6.07%) |
| Focal loss | 83.92 (18.66%) | 90.41 (9.17%) | 96.07 (3.28%) | 79.24 (25.83%) |
| DNN-EB Bagging | 97.31 (2.33%) | 96.11 (2.69%) | 95.77 (3.60%) | 92.10 (8.26%) |
| HHO-OIF | 96.89 (2.78%) | 96.31 (2.48%) | 96.88 (2.41%) | 96.22 (3.63%) |
| cost-sensitive GA-DNN-RB Bagging | 96.74 (2.94%) | 94.66 (4.27%) | 83.78 (18.43%) | 88.30 (12.92%) |
| cost-sensitive PSO-DNN-RB Bagging | 97.44 (2.20%) | 95.79 (3.04%) | 88.10 (12.62%) | 90.20 (10.54%) |
| cost-sensitive ABC-DNN-RB Bagging | 97.81 (1.81%) | 96.60 (2.17%) | 89.76 (10.54%) | 90.68 (9.96%) |
| Proposed | 99.58 | 98.70 | 99.22 | 99.71 |

exhibits a consistent performance for all datasets. These results demonstrate the excellent ability of our IDS to detect attacks in the four intrusion detection datasets.

Further, the F1-score metric is investigated as it serves as the harmonic mean of the precision and recall metrics. It can be inferred from Table 6 that our IDS achieves higher values of the F1-score metric for all the four datasets as compared to other IDSs. Specifically, the F1-score metric for our IDS lies within 98.70–99.71%. When comparing

the proposed model with the recently-proposed ones, an improvement in the F1-score of order of about 2% to 3% over the best model is observed. When comparing our FPA-based model with other metaheuristics-based models, we notice an improvement in the F1-score of order of about 2% to 11% over the ABC-based IDS. Also, we observe that other metaheuristics-based models perform slightly worse for the CIC-IDS-2017 dataset. Due to the large number of features in this dataset compared to the other datasets, the performance of GA, PSO, and ABC techniques is considerably degraded.

The above results reveal that the proposed method consistently achieves optimal performance for imbalanced datasets in terms of the accuracy, precision, recall, and F1-score metrics. This can be attributed to the integration of cost-sensitive objective function, FPA, DNN, and ensemble learning. First, the use of a cost-sensitive objective function enhances the learning capacity of the individual DNN models with regard to the minority class. Second, the global search capability of FPA enables accurate and diverse base learners required for an effective ensemble learning. Third, the ensemble learning alleviates the over-predicting toward the minority class and rectifies for the individual learners' misleading prediction, and hence improves the generalization performance of the designed IDS for unseen data.

As a final note on the limitations of this study, we did not take into account integrating any defense mechanism for our IDS against adversarial attacks, which are perturbed inputs constructed to fool ML/DL-based IDSs. From input space point of view, adversarial attack can be feature space or problem space (*Ibitoye et al., 2019*). The former manipulates the features of an instance directly, and does not generate a new instance. The latter tends to create a new object by modifying the actual instance itself. In general, the resilience and robustness of the proposed IDS can be improved using existing defense strategies such as adversarial training, feature reduction, input randomization, and ensemble defenses (*Ibitoye et al., 2019*). However, it should be noted that, in contrast to computer vision that relies on images, network traffic consists of data objects, therefore the perturbed features are more diverse and heterogeneous. As a result, adversarial attacks pose a more significant threat in network security.

## CONCLUSION

This article has proposed a new intrusion detection system based on a hybridization of a cost-sensitive metaheuristic-deep learning model with an ensemble learning method. A deep neural network, whose parameters are optimized by flower pollination algorithm on a cost-sensitive fitness function, is utilized as a base learner in a roughly-balanced Bagging strategy. Each base learner is trained using a unique roughly-balanced training subset derived from the original imbalanced training set, where an appropriate class weight is used in the objective function. The performance of the proposed method is evaluated on four widely-known datasets, NSL-KDD, UNSW NB15, CIC-IDS-2017, and BoT-IoT, in terms of the accuracy, precision, recall, and F1-score metrics. Comparisons with IDSs based on widely-used metaheuristics, genetic algorithm, particle swarm optimization, and artificial bee colony, as well as recently-proposed techniques were conducted. The results demonstrate that the proposed method consistently outperforms all the others for the four

datasets, underscoring its potential in handling the class imbalance classification problem in intrusion detection datasets. We found that the cost-sensitive objective function enhances the learning capacity with regard to the minority class, whereas the ensemble learning alleviates the over-predicting toward the minority class and rectifies for the individual learners' misleading prediction. Yet, the global search capability of FPA enables accurate and diverse base learners required for an effective ensemble learning. The implication of our findings is that, integrating the proposed technique into real-world deployment in network security applications could enhance the detection capabilities of intrusion detection systems. The proposed method can be established as a cutting-edge intrusion detection solution, providing robust defense against evolving cyber threats in IoT infrastructures.

Future work can be devoted to investigate the performance of the proposed method in other intrusion detection datasets as well as other class imbalance classification problems such as medical diagnosis. While our current evaluation demonstrates strong performance on established IDS datasets, future work could explore the method's effectiveness on more severely imbalanced scenarios, such as those found in APT detection where attack samples may constitute as little as 0.04% of the data (*Benabderrahmane et al., 2024*). Such an evaluation would help validate the approach's robustness under extreme class imbalance conditions. Another research direction of interest is to integrate our framework with metaheuristic-based feature selection and hyper-parameters tuning.

### Funding
The authors received no funding for this work.

### Competing Interests
The authors declare that they have no competing interests.

### Author Contributions
- Hussein Ridha Sayegh conceived and designed the experiments, performed the experiments, analyzed the data, performed the computation work, prepared figures and/or tables, authored or reviewed drafts of the article, and approved the final draft.
- Wang Dong conceived and designed the experiments, performed the experiments, prepared figures and/or tables, authored or reviewed drafts of the article, and approved the final draft.
- Bahaa Hussein Taher analyzed the data, performed the computation work, authored or reviewed drafts of the article, and approved the final draft.
- Muhanad Mohammed Kadum analyzed the data, performed the computation work, prepared figures and/or tables, and approved the final draft.
- Ali Mansour Al-madani conceived and designed the experiments, performed the experiments, analyzed the data, performed the computation work, prepared figures and/or tables, authored or reviewed drafts of the article, and approved the final draft.

## Data Availability

The raw data and code are available in the Supplemental Files.

## Supplemental Information

Supplemental information for this article can be found online at http://dx.doi.org/10.7717/peerj-cs.2745#supplemental-information.

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
