# Peer review of "Optimal intrusion detection for imbalanced data using Bagging method with deep neural network optimized by flower pollination algorithm"

_PeerJ Computer Science, doi:10.7717/peerj-cs.2745_

## Round 0.1 · original submission · Major Revisions

Dear authors,

Thank you for the submission. The reviewers’ comments are now available. It is not suggested that your article be published in its current format. We do, however, advise you to revise the paper in light of the reviewers’ comments and concerns before resubmitting it. The followings should also be addressed:

1. The real contributions should be listed in the manuscript. The performed works are shown as contributions. Using metaheuristics for determining the weight distribution of neural networks are used and how your approach is original is not discussed.
2. The motivation and reason of using Flower Pollination Algorithm among many other metaheuristic algorithms for optimizing weights of deep neural network is not mentioned.
3. All of the values for the parameters of all algorithms should be provided.
4. Literature review section and performance evaluation and comparisons section are weak.
5. Encoding or representation scheme and fitness function of Flower Pollination Algorithm should be provided.
6. How constraints for Flower Pollination Algorithm (for example in decision variables intervals) are handled is not clear.
7. Many of the equations are part of the related sentences. Attention is needed for correct sentence formation.
8. Equations should be used with correct equation number. Please do not use “as follows”, “given as”, etc. Explanation of the equations should also be checked. All variables should be written in italic as in the equations. Their definitions and boundaries should be defined. Necessary references should be provided.
9. Some more recommendations and conclusions should be discussed about the paper considering the experimental results. The conclusion section is weak. There is also no discussion section about the results. It should briefly describe the results of the study and some more directions for further research. You should describe the academic implications, main findings, shortcomings and directions for future research in the conclusion section. The conclusion in its current form is generally confused. What will be happen next? What we supposed to expect from the future papers? So rewrite it and consider the following comments:
- Highlight your analysis and reflect only the important points for the whole paper.
- Mention the benefits
- Mention the implication in the last of this section.

Best wishes,

Reviewer 1 ·

Basic reporting

The paper is well written but here are some concerns:

The paper discusses an intrusion detection system using imbalances datasets. As any classical imbalanced anomaly detection and classification problem, the authors used a data augmentation technique.
The big concern in this strategy is how to verify the sanity and the quality of the generated synthetic data in the context of network traffic and cyber attacks. Unlike images or videos (when using GANS or any other generative model, ... ) where we could verify the veracity the content visually; how could we check the semantics in the generated sequences in this context. Many authors have dealt in the past with adversarial attacks for IDS. I think the author should have a look at these works (from feature space, and problem space point of view).

It is not clear how a tiny class of true positives (even though augmented) could be used to train a DNN.
Authors need to clearly explain this point.

Experimental design

Non supervised methods are widely more adapted to these kind of imbalanced classification problem.
The authors should have compared there work with other methods such as isolation forests or oc-svm

Validity of the findings

ND

Additional comments

types

line 173: multiplecbase

Reviewer 2 ·

Basic reporting

The paper is well written but there are some sentences are complex and could be simplified for better readability. For example at line 164 ."The DNN is optimized by using FPA ....data sets". Recommend to proofread the paper to correct grammatical errors and improve sentence structure.

There is some redundancy in the content of the introduction and literature review sections. For example the literature review mentions various machine learning (ML), deep learning (DL) and metaheuristic algorithms which were already stated in the introduction section. The discussion for the requirement of performance improvement also been repeated in both sections (check line 67-70 and line 148-152.Thus, need to refine the structure for the introduction and literature review section.
Suggest to add background information on Intrusion Detection Systems (IDS) and discussion of common approaches used in IDS, particularly focusing on the issue of imbalanced datasets, DL and hybrid model. This will highlight the issues that the paper aims to address and the novelty of study
Refine the literature review section by critically review previous works related on IDS including deep learning, hybrid metaheuristic algorithms with deep learning method and approaches for handling class imbalance. Provide more specific details about the performance, strengths and limitations of existing method to identify the gap especially. the benchmark method used in the analysis. Update references to ensure they are relevant and recent, with a focus on work published within the last 5 years.

The sections in the paper are not numbered then need to use a more general approach to describe the structure of the paper without numbering in line 105-108.

To improve the flow and readability of the paper, ensure that the figures, tables, and algorithms are positioned close to the text it been cited and make sure it been referenced correctly in the text .For example :
- Fig 3 at Line 239 should be replaced with Fig 2 as it refer to Figure 2.
-Missing citation for Algorithm 2 and suggest to cite at line 236.
-Table 2 should be place within/after DNN Architecture section.
- Rearrange Figure 3 until Figure 10 to be placed within the Performance Evaluation and Comparison section.
-Cite all tables in text using Arabic numerals (1,2,3) instead of Roman numerals(I,II,III). Check the citation for the Table 1 until Table 6 in the paper.

Suggest to revise the structure of the paper from understanding the context (Introduction, Literature review) to the methodology (Methods) followed by the proposed method and its evaluation.

The results for each dataset have been provided, but it would be more useful to include the raw datasets. Add information on the number of benign and attack for each dataset in Table 1 to illustrate the class imbalance and provide general discussion on charasteristics element for the datasets.

Experimental design

-Suggest to restructure the flow of the paper. The "Proposed Model" section currently presents before the "Experiment Setup" section. It would be more appropriate to provide the framework or method of the proposed IDS model before discussing the specific more details on the proposed model.
-The title for the “Experiment Setup” section would be more appropriate to be rename as “ Methods” to better represent the overall phase for conducting the study including data sets, data pre processing and evaluation metricsr than just focusing on the experimental design for the model.
-Check F1 formula in equation 10
-Provide justification of parameter used in DNN architecture including the 3 number of hidden layer, activation function( Relu and sigmoid), size of neuron and batch size.
- Discuss the data splitting method for the study.

Validity of the findings

-Revise content for column method in Table 2 until Table 6 to remove citations and include the benchmark methods as presented in the literature review section.
- The results presented in the 'Performance Evaluation and Comparisons' section are inconsistent with the results shown in the tables. For example, in Table 3, the proposed model shows the lowest accuracy value as 99.32% for the CIC-IDS 2017 dataset and the highest accuracy is 99.59% for the NSL-KDD dataset. However, at line 304, it is stated that 'the model lies within 98.34-99.59%.' Please provide a more detailed explanation of these discrepancies and add also percentage of improvement compared with the benchmark model.
-Provide discussion of the result presented in the confusion matrix and loss function figures.
-In the conclusion section, highlight the impact of the study and outline the direction for the future work.

Additional comments

no

---

## Round 0.2 · Minor Revisions

Dear Authors,

It is strongly recommended that the concerns of Reviewer 1 be addressed, and that the criticisms and feedback be taken on board. Once the article has been updated, it should be resubmitted. In addition, it is necessary to ensure that equations are used with the correct equation number. Many of the equations are part of the related sentences, and care must be taken to ensure correct sentence formation.

Best wishes,

Reviewer 1 ·

Basic reporting

The general shape of the paper is good, but the quality of the figures needs to be improved.

Experimental design

There is a lack of comparison with recent research papers in the area.
It is not clear how ensemble learning impacts the data distribution and its "imbalance", i.e, what is the proportion of the attacks during bagging.

Validity of the findings

I suggest not to only focus on NSL-KDD, UNSW NB-15, CIC-IDS-2017, and BoT-IoT because these are not purely imbalanced. There are a lot of other datasets in cybersecurity dealing with advanced persistent threats which are very imbalanced :
For instance you've got:
https://www.sciencedirect.com/science/article/pii/S0167739X24003479?via%3Dihub


The datasets in the previous paper are extremely imbalanced (attacks represent 0.04%) hence applying your approach is worth it.

---

## Round 0.3 · accepted · Accept

Dear Authors,

Thank you for addressing the reviewers' comments. Your manuscript now seems sufficiently improved and ready for publication.

Best wishes,

Reviewer 1 ·

Basic reporting

Major previous points have been included.

Experimental design

Please enhance the quality of figure 2

Validity of the findings

Authors are encouraged to include some information about running time and complexity.